# Observer-Based Adaptive Decentralized Control for Interconnected Nonlinear Systems with Input Delay

Qixia Shen
*College of Mathematical Sciences*
*Bohai University*
Jinzhou, China
shenqixia0219@163.com

Yushan Cen
*College of Mathematical Sciences*
*Bohai University*
Jinzhou, China
cenyushan2023@163.com

Liang Cao
*College of Mathematical Sciences*
*Institute of Ocean Research*
*Bohai University*
Jinzhou, China
caoliang0928@163.com

*Abstract*—For nonlinear interconnected systems against input delay, an observer-based fuzzy adaptive decentralized control strategy is designed. First, different from traditional observers, a novel state observer based on tracking error is proposed to estimate unknown states for the interconnected systems. Secondly, a compensation signal is added to handle the time-varying input delay. Then, the dynamic surface control techniques are utilized in the backstepping approach to overcome the "explosion of complex" problem. Meanwhile, through designed schemes, all variables in the closed-loop systems are promised to be bounded. The errors converge into bounded compact set around the origin. The effectiveness of the designed scheme is validated via a simulation example.

*Index Terms*—Dynamic surface control, nonlinear interconnected systems, state observer, time-varying input delay.

## I. Introduction

In the past decade, interconnected systems are typically used to depict complex systems composed of multiple subsystems that have physical couplings or network interactions. Due to their large scale, the decentralized control based on backstepping is widely applied in nonlinear interconnected systems. In [1], the authors put forward an adaptive fuzzy decentralized control strategy for interconnected systems. In the derivation process of the backstepping method, it is necessary to differentiate the virtual controllers, which leads to the computational complexity problem. To overcome "explosion of complexity" problem, a dynamic surface control (DSC) method was put forward by performing first-order filters at each step. Xu *et al.* [2] designed a predefined time filter to address "explosion of complex" issue. The decentralized DSC scheme was designed to avoid the repetitive derivative of virtual controllers for large-scale power systems [3]. In practical engineering, the stability of the systems are also affected by many factors, such as input delay and unmeasurable system state.

Due to limited equipment, nonstandard operations, environmental disturbances and signal transmission, input delay often occurs in practical systems, which may be the source of unstable factors that lead to unexpected degradation of the system. Therefore, how to solve the input delay has become a key issue. In [4], the input delay was solved by Pade approximation. Xing *et al.* [5] constructed an auxiliary system to offset the unfavorable effects of input delay. The authors in [6] overcame the influence of the input delay by proposing a compensation signal. In practical application, state variables of interconnected systems are generally unmeasurable. Specially, observers are often adopted to solve this problem [7]. Xu *et al.* [8] proposed a state observer based on fuzzy logic systems (FLSs) for interconnected systems. A finite-dimensional observer was constructed on the basis of system output to asymptotically estimate state information [9]. In [10], the authors constructed a distributed state observer based on synchronization error for multiagent systems to estimate unmeasured states. Many achievements about observers of interconnected systems based on output information. Inspired by distributed observers, it is significant to design an observer based on tracking errors to estimate unknown states for large-scale nonlinear systems against input delay.

Motivated by the aforementioned discussions, this paper puts forward an observer-based adaptive decentralized control method for nonlinear interconnected systems against input delay. The main contributions are described below.

1) An observer based on tracking errors is investigated for interconnected systems with input delay to estimate unmeasured states. On the basis of traditional state observers, not only the output variables but also the information of the reference signal are considered.

2) With the framework of backstepping control, a compensation signal is added to deal with input delay, and the system stability is assured. In order to decrease the computational burden, DSC and first-order filters are adopted to handle complexity explosion problems.

## II. Preliminaries

The interconnected systems against time-varying input delay are modeled as follows:

$$\begin{cases} \dot{x}_{i,m} = x_{i,m+1} + \Delta_{i,m}(\bar{y}) + f_{i,m}(\bar{x}_{i,m}) \\ \dot{x}_{i,n} = u_i(t - \tau_i(t)) + \Delta_{i,n_i}(\bar{y}) + f_{i,n_i}(\bar{x}_{i,n_i}) \\ y_i = x_{i,1}, \ i = 1, \ldots, N, \ m = 1, \ldots, n_i - 1 \end{cases} \quad (1)$$

where $\bar{x}_{i,m} = [x_{i,1}, \ldots, x_{i,m}]^T \in \mathbb{R}^m$ and $\bar{x}_{i,n_i} = [x_{i,1}, \ldots, x_{i,n_i}]^T \in \mathbb{R}^{n_i}$ are system states. $u_i \in \mathbb{R}^N$ and $y_i \in \mathbb{R}^N$ denote the system input and output, respectively. $\tau_i(t)$ represents a known time-varying function. $\Delta_{i,m}(\bar{y})$ and $\Delta_{i,n_i}(\bar{y})$ mean unknown smooth interconnections, which

$\bar{y} = [y_1, \ldots, y_N]$. $f_{i,m}(\bar{x}_{i,m})$ and $f_{i,n_i}(\bar{x}_{i,n_i})$ are unknown smooth nonlinear functions. It is assumed that only output $y_i$ can be obtained.

*Assumption 1:* For two positive constants $\bar{\tau}_i$ and $\hbar_i$, the input delay $\tau_i(t)$ contents $\tau_i(t) \leq \bar{\tau}_i$ and $\dot{\tau}_i(t) \leq \hbar_i < 1$.

*Assumption 2:* A constant $\iota_{i,m}$ satisfies the inequality

$$|\Delta f_{i,m}| \leq \iota_{i,m}\|\bar{x}_{i,m} - \hat{\bar{x}}_{i,m}\|$$

where $\Delta f_{i,m} = f_{i,m}(\bar{x}_{i,m}) - f_{i,j}(\hat{\bar{x}}_{i,m})$. $\hat{\bar{x}}_{i,m} = [\hat{x}_{i,1}, \ldots, \hat{x}_{i,m}]^T$ is the estimate of $\bar{x}_{i,m}$.

*Assumption 3:* For $m = 1, \ldots, n_i$, there is an unknown smooth function $\rho_{i,m,p}(\cdot)$ satisfying

$$|\Delta_{i,m}(\bar{y})|^2 \leq \sum_{p=1}^{N} \rho_{i,m,p}^2(y_p)$$

*Lemma 1:* Let $f(x)$ on the compact set $\Omega$ be the continuous function. There is an FLS $y(x) = \Theta^T \varphi(x)$ for any constant $\delta > 0$ satisfying

$$\sup_{x \in \Omega} |f(x) - \Theta^T \varphi(x)| \leq \delta$$

In the light of Lemma 1, the unknown smooth function in (1) can be approximated by the following FLSs:

$$\hat{f}_{i,m}(\hat{\bar{x}}_{i,m}|\Theta_{i,m}) = \Theta_{i,m}^T \varphi_{i,m}(\hat{\bar{x}}_{i,m})$$

Define the optimal parameter vectors $\Theta_{i,m}^*$ as

$$\Theta_{i,m}^* = \arg \min_{\Theta_{i,m}^* \in \Omega_{i,m}} [\sup_{\hat{\bar{x}}_{i,m} \in \hat{U}_{i,m}} |\hat{f}_{i,m}(\hat{\bar{x}}_{i,m}|\Theta_{i,m}) - f_{i,m}(\hat{\bar{x}}_{i,m})|]$$

in which the compact regions of $\Theta_{i,m}^*$ and $\hat{\bar{x}}_{i,m}$ are showed as $\Omega_{i,m}$ and $\hat{U}_{i,m}$, respectively.

The following fuzzy minimum approximation errors $\delta_{i,m}$ and approximation errors $\omega_{i,m}$ are defined:

$$\delta_{i,m} = f_{i,m}(\hat{\bar{x}}_{i,m}) - \hat{f}_{i,m}(\hat{\bar{x}}_{i,m}|\Theta_{i,m}^*)$$
$$\omega_{i,m} = f_{i,m}(\hat{\bar{x}}_{i,m}) - \hat{f}_{i,m}(\hat{\bar{x}}_{i,m}|\Theta_{i,m})$$

*Assumption 4:* Two known constants $\delta_{i,m}^* > 0$ and $\omega_{i,m}^* > 0$ fulfill $|\delta_{i,m}| \leq \delta_{i,m}^*$ and $|\omega_{i,m}| \leq \omega_{i,m}^*$.

*Lemma 2:* There exist positive real numbers $k_{i,1}, \ldots, k_{i,n_i}$, Hurwitz matrix $A_i$, $P_i = P_i^T > 0$ and $\varsigma_i = \varsigma_i^T > 0$ fulfilling

$$A_i^T P_i + P_i A_i \leq -\varsigma_i$$

where

$$A_i = \begin{bmatrix} -k_{i,1} & & \\ \vdots & & I_{(n_i-1)\times(n_i-1)} \\ -k_{i,n_i-1} & & \\ -k_{i,n_i} & \cdots & 0 \end{bmatrix}.$$

## III. MAIN RESULTS

### A. *Tracking Error-Based Observer Design*

Due to the unavailability of the states in (1), an observer based on the tracking error is designed

$$\begin{cases} \dot{\hat{x}}_{i,m} = \hat{x}_{i,m+1} + \hat{f}_{i,m}(\hat{\bar{x}}_{i,m}|\Theta_{i,m}) + k_{i,m}e_{i,1} \\ \dot{\hat{x}}_{i,n_i} = u_i(t - \tau_i(t)) + \hat{f}_{i,n}(\hat{\bar{x}}_{i,n_i}|\Theta_{i,n_i}) + k_{i,n_i}e_{i,1} \end{cases} \quad (2)$$

where $e_{i,1}$ expresses the tracking error, which will be provided later.

Define the following observation error:

$$\varepsilon_{i,m} = x_{i,m} - \hat{x}_{i,m} \quad (3)$$

From (2) and (3), one has

$$\dot{\varepsilon}_i = A_i \varepsilon_i + K_i L_i \varepsilon_i + \sum_{m=1}^{n_i} B_{i,m}(\Delta f_{i,m} + \omega_{i,m} + \Delta_{i,m})$$
$$- K_i e_{i,1} \quad (4)$$

where $L_i = [1, 0, \ldots, 0]$ and $B_{i,m} = [\underbrace{0, \ldots, 1}_{m}, \ldots, 0]^T$.

The following Lyapunov function is considered:

$$V_{i,0} = \varepsilon_i^T P_i \varepsilon_i \quad (5)$$

In view of the above equation, one gets

$$\dot{V}_{i,0} \leq -\varepsilon_i^T \varsigma_i \varepsilon_i + 2\varepsilon_i^T P_i K_i^T L_i^T \varepsilon_i - 2\varepsilon_i^T P_i K_i e_{i,1}$$
$$+ 2\varepsilon_i^T P_i \sum_{m=1}^{n_i} B_{i,m}(\omega_{i,m} + \Delta_{i,m} + \Delta f_{i,m}) \quad (6)$$

With Young's inequality, Assumptions 2-4, one has

$$2\varepsilon_i^T P_i \sum_{m=1}^{n_i} B_{i,m} \Delta f_{i,m} \leq \lambda_{\max}^2(P_i)\|\varepsilon_i\|^2 + \sum_{m=1}^{n_i} \iota_{i,m}^2\|\varepsilon_i\|^2 \quad (7)$$

$$2\varepsilon_i^T P_i \sum_{m=1}^{n_i} B_{i,m}\omega_{i,m} \leq \lambda_{\max}^2(P_i)\|\varepsilon_i\|^2 + \|\omega_i^*\|^2 \quad (8)$$

$$2\varepsilon_i^T P_i \sum_{m=1}^{n_i} B_{i,m}\Delta_{i,m} \leq \lambda_{\max}^2(P_i)\|\varepsilon_i\|^2 + \sum_{p=1}^{N}\sum_{m=1}^{n_i} \rho_{i,m,p}^2(y_p)$$
$$= \lambda_{\max}^2(P_i)\|\varepsilon_i\|^2 + y_i^2 \sum_{p=1}^{N}\sum_{m=1}^{n_p} \bar{\rho}_{p,m,i}^2(y_i) \quad (9)$$

$$2\varepsilon_i^T P_i K_i e_{i,1} \leq \varepsilon_i^T P_i K_i K_i^T P_i \varepsilon_i + e_{i,1}^2 \quad (10)$$

where $\omega_i^* = [\omega_{i,1}^*, \ldots, \omega_{i,n_i}^*]^T$. $\bar{\rho}_{p,m,i}(y_i)$ means an unknown smooth function.

By substituting (7)-(10) into (6), one obtains

$$\dot{V}_{i,0} \leq -\left(\lambda_{\min}(\varsigma_i) - 2\lambda_{\max}(P_i K_i L_i) - \lambda_{\max}(P_i K_i K_i^T P_i)\right.$$
$$\left. - 3\lambda_{\max}^2(P_i) - \sum_{m=1}^{n_i} \iota_{i,m}^2\right)\|\varepsilon_i\|^2 + y_i^2 \sum_{p=1}^{N}\sum_{m=1}^{n_p} \bar{\rho}_{p,m,i}^2(y_i)$$
$$+ \|\omega_i^*\|^2 + e_{i,1}^2 \quad (11)$$

## B. Decentralized Control Design

To decrease the computational burden, the first-order filter is added, and the coordinate transformation is given

$$\begin{cases} e_{i,1} = y_i - y_{i,d} \\ e_{i,j} = \hat{x}_{i,j} - \bar{\alpha}_{i,j-1}, \ j = 2,\ldots,n_i-1 \\ e_{i,n_i} = \hat{x}_{i,n_i} - \bar{\alpha}_{i,n_i-1} + \eta_i \\ e_{\alpha_{i,j-1}} = \bar{\alpha}_{i,j-1} - \alpha_{i,j-1} \\ e_{\alpha_{i,n_i}} = \bar{\alpha}_{i,n_i} - \alpha_{i,n_i} \end{cases} \tag{12}$$

where $\alpha_{i,j-1}$ and $\alpha_{i,n_i}$ stand for the virtual controllers. $\bar{\alpha}_{i,j-1}$ and $\bar{\alpha}_{i,n_i}$ indicate the first-order filters, and $e_{\alpha_{i,j-1}}$ and $e_{\alpha_{i,n_i}}$ show the output errors of filters. The auxiliary signal $\eta_i$ will be provided later.

For fear of repeated differentiation of virtual controllers, this paper introduces a first-order filters $\bar{\alpha}_{i,j-1}$ and $\bar{\alpha}_{i,n_i}$, and one has

$$\begin{cases} \sigma_{i,j-1}\dot{\bar{\alpha}}_{i,j-1} + \bar{\alpha}_{i,j-1} = \alpha_{i,j-1} \\ \sigma_{i,n_i}\dot{\bar{\alpha}}_{i,n_i} + \bar{\alpha}_{i,n_i} = \alpha_{i,n_i} \\ \bar{\alpha}_{i,j-1}(0) = \alpha_{i,j-1}(0) \\ \bar{\alpha}_{i,n_i}(0) = \alpha_{i,n_i}(0) \end{cases} \tag{13}$$

where $\sigma_{i,j-1}$ and $\sigma_{i,n_i}$ are positive constants. Based on (12) and (13), it yields that

$$\begin{cases} \dot{e}_{\alpha_{i,j-1}} = -\dfrac{e_{\alpha_{i,j-1}}}{\sigma_{i,j-1}} + O_{i,j-1}(\cdot) \\ \dot{e}_{\alpha_{i,n_i}} = -\dfrac{e_{\alpha_{i,n_i}}}{\sigma_{i,n_i}} + O_{i,n_i}(\cdot) \end{cases} \tag{14}$$

where $O_{i,j-1}(\cdot) = -\dot{\alpha}_{i,j-1}$ and $O_{i,n_i}(\cdot) = -\dot{\alpha}_{i,n_i}$ denote the continuous functions.

**Step $i$, 1**: It follows from (1), (2) and (12) that

$$\dot{e}_{i,1} = e_{i,2} + e_{\alpha_{i,1}} + f_{i,1}(\bar{x}_{i,1}) + \alpha_{i,1}$$
$$+ \Delta_{i,1} + \varepsilon_{i,2} - \dot{y}_{i,d} \tag{15}$$

Construct the Lyapunov function as

$$V_{i,1} = V_{i,0} + \frac{1}{2}e_{i,1}^2 + \frac{1}{2}e_{\alpha_{i,1}}^2 + \frac{1}{2\Upsilon_{i,1}}\tilde{\Theta}_{i,1}^2 + \frac{1}{2\zeta_{i,1}}\tilde{W}_i^2 \tag{16}$$

where $\tilde{\Theta}_{i,1} = \Theta_{i,1}^* - \Theta_{i,1}$ and $\tilde{W}_i = W_i^* - W_i$. $\Theta_{i,1}$ and $W_i$ denote the estimations of $\Theta_{i,1}^*$ and $W_i^*$, respectively. $W_i^*$ will be explained later. $\Upsilon_{i,1}$ and $\zeta_{i,1}$ are both designed constants.

In view of (15) and (16), the derivative of $V_{i,1}$ is

$$\dot{V}_{i,1} = \dot{V}_{i,0} + e_{i,1}\left(\varepsilon_{i,2} + e_{i,2} + e_{\alpha_{i,1}} + f_{i,1}(\bar{x}_{i,1}) - \dot{y}_{i,d}\right.$$
$$\left. + \alpha_{i,1} + \Delta_{i,1}\right) + e_{\alpha_{i,1}}\left(-\frac{e_{\alpha_{i,1}}}{\sigma_{i,1}} + O_{i,1}\right)$$
$$- \tilde{\Theta}_{i,1}^T \Upsilon_{i,1}^{-1}\dot{\Theta}_{i,1} - \tilde{W}_i^T \zeta_{i,1}^{-1}\dot{W}_i \tag{17}$$

According to the Young's inequality, Lemma 1 and Assumption 4, one has

$$e_{i,1}(e_{\alpha_{i,1}} + e_{i,2} + \varepsilon_{i,2}) \leq \frac{3}{2}e_{i,1}^2 + \frac{e_{\alpha_{i,1}}^2 + e_{i,2}^2 + \|\varepsilon_i\|^2}{2} \tag{18}$$

$$e_{\alpha_{i,1}}O_{i,1} \leq \frac{1}{2q_{i,1}^2}e_{\alpha_{i,1}}^2 O_{i,1}^2 + \frac{1}{2}q_{i,1}^2 \tag{19}$$

$$e_{i,1}f_{i,1}(\bar{x}_{i,1}) \leq \frac{3}{4}e_{i,1}^2 + \frac{1}{2}\delta_{i,1}^{*2} + e_{i,1}\tilde{\Theta}_{i,1}^T\varphi_{i,1}(\hat{\bar{x}}_{i,1})$$
$$+ \iota_{i,1}^2\|\varepsilon_i\|^2 + e_{i,1}\Theta_{i,1}^T\varphi_{i,1}(\hat{\bar{x}}_{i,1}) \tag{20}$$

$$e_{i,1}\Delta_{i,1} \leq \frac{1}{2}e_{i,1}^2 + y_i^2\sum_{p=1}^{N}\bar{\rho}_{p,1,i}^2(y_i) \tag{21}$$

in which the constant $q_{i,1}$ is positive.

This paper applies the FLS $\hat{b}_i(y_i|W_i) = W_i^T\phi_i(y_i)$ to approximate the interconnection terms $y_i\sum_{p=1}^{N}\sum_{m=1}^{n_p}\bar{\rho}_{p,m,i}^2(y_i)$ and $y_i\sum_{p=1}^{N}\bar{\rho}_{p,1,i}^2(y_i)$ in (11) and (21).

Define the optimal vector $W_i^*$ as

$$W_i^* = \arg\min_{W_i^*\in\Pi_i}\left[\sup_{y_i\in\Psi_i}\left|\hat{b}_i(y_i|W_i) - \left[y_i\sum_{p=1}^{N}\sum_{m=1}^{n_p}\bar{\rho}_{p,m,i}^2(y_i)\right.\right.\right.$$
$$\left.\left.\left. + y_i\sum_{p=1}^{N}\bar{\rho}_{p,1,i}^2(y_i)\right]\right|\right] \tag{22}$$

where $\Pi_i$ and $\Psi_i$ are compact regions for $W_i^*$ and $y_i$, respectively. Define the minimum approximation error $\xi_i$ as $\xi_i = y_i\sum_{p=1}^{N}\sum_{m=1}^{n_p}\bar{\rho}_{p,m,i}^2(y_i) + y_i\sum_{p=1}^{N}\bar{\rho}_{p,1,i}^2(y_i) - \hat{b}_i(y_i|W_i^*)$. There is a known positive constant $\xi_i^*$ such that $|\xi_i| \leq \xi_i^*$.

Based on the above equation, one gets

$$y_i^2\sum_{p=1}^{N}\sum_{m=1}^{n_p}\bar{\rho}_{p,m,i}^2(y_i) + y_i^2\sum_{p=1}^{N}\bar{\rho}_{p,1,i}^2(y_i)$$
$$\leq \frac{1}{2}e_{i,1}^2 + \frac{1}{2}W_i^{*2} + e_{i,1}W_i^*\phi_i(y_i) + \xi_i^{*2} + y_{i,d}^2 \tag{23}$$

Further, we obtain

$$\alpha_{i,1} = -c_{i,1}e_{i,1} - W_i^T\phi_i(y_i) - \Theta_{i,1}^T\varphi_{i,1}(\hat{\bar{x}}_{i,1})$$
$$+ \dot{y}_{i,d} - \frac{17}{4}e_{i,1} \tag{24}$$

$$\dot{\Theta}_{i,1} = \Upsilon_{i,1}(e_{i,1}\varphi_{i,1}(\hat{\bar{x}}_{i,1}) - \gamma_{i,1}\Theta_{i,1}) \tag{25}$$

$$\dot{W}_i = \zeta_i(e_{i,1}\phi_i(y_i) - l_iW_i) \tag{26}$$

in which $\gamma_{i,1} > 0$ and $l_i > 0$ are both designed parameters.

Based on the above analysis, $\dot{V}_{i,1}$ is transformed into

$$\dot{V}_{i,1} \leq -\mu_i\|\varepsilon_i\|^2 - c_{i,1}e_{i,1}^2 + \frac{1}{2}e_{i,2}^2 + \|\omega_i^*\|^2 + \frac{1}{2}q_{i,1}^2$$

$$e_{\alpha_{i,1}}^2 + \frac{1}{2}\delta_{i,1}^{*2} + \xi_i^{*2} - \left(\frac{1}{\sigma_{i,1}} - \frac{1}{2q_{i,1}^2}O_{i,1}^2 - \frac{1}{2}\right)$$

$$+ \frac{1}{2}W_i^{*2} + y_{i,d}^2 + \tilde{\Theta}_{i,1}^T\gamma_{i,1}\Theta_{i,1}^T + \tilde{W}_{i,1}^Tl_iW_{i,1}^T \tag{27}$$

in which $\mu_i = \lambda_{\min}(\varsigma_i) - 2\lambda_{\max}(P_iK_iL_i) - 3\lambda_{\max}^2(P_i)\iota_{i,1}^2 - \lambda_{\max}(P_iK_iK_i^TP_i) - \sum_{m=1}^{n_i}\iota_{i,m}^2 - \iota_{i,1}^2 - \frac{1}{2}$.

**Step $i, j$ ($2 \leq j \leq n-1$):** The expression of the Lyapunov function is

$$V_{i,j} = V_{i,j-1} + \frac{1}{2}e_{i,j}^2 + \frac{1}{2\Upsilon_{i,j}}\tilde{\Theta}_{i,j}^2 + \frac{1}{2}e_{\alpha_{i,j}}^2 \qquad (28)$$

where $\Upsilon_{i,j}$ expresses a normal number. $\Theta_{i,j}$ denotes the estimation of $\Theta_{i,j}^*$, and the error between them is $\tilde{\Theta}_{i,j} = \Theta_{i,j}^* - \Theta_{i,j}$.

In the light of (12), the differentiation of $V_{i,j}$ is calculated as

$$\dot{V}_{i,j} = e_{i,j}\Big(k_{i,j}e_{i,1} + e_{i,j+1} + \tilde{\Theta}_{i,j}^T\varphi_{i,j}(\hat{\bar{x}}_{i,j}) + \alpha_{i,j} + e_{\alpha_{i,j}}$$
$$+ \delta_{i,j} + \Theta_{i,j}^T\varphi_{i,j} - \omega_{i,j} - \dot{\alpha}_{i,j-1}\Big) + \dot{V}_{i,j-1}$$
$$+ e_{\alpha_{i,j}}\left(-\frac{e_{\alpha_{i,j}}}{\sigma_{i,j}} + O_{i,j}\right) - \tilde{\Theta}_{i,j}^T\Upsilon_{i,j}^{-1}\dot{\Theta}_{i,j} \qquad (29)$$

Based on the Young's inequality, one has

$$e_{i,j}(e_{i,j+1} + e_{\alpha_{i,j}}) \leq e_{i,j}^2 + \frac{1}{2}e_{\alpha_{i,j}}^2 + \frac{1}{2}e_{i,j+1}^2 \qquad (30)$$

$$e_{\alpha_{i,j}}O_{i,j} \leq \frac{1}{2}q_{i,j}^2 + \frac{1}{2q_{i,j}^2}e_{\alpha_{i,j}}^2 O_{i,j}^2 \qquad (31)$$

$$e_{i,j}(\omega_{i,j} + \delta_{i,j}) \leq \frac{1}{2}(\delta_{i,j}^{*2} + \omega_{i,j}^{*2}) + e_{i,j}^2 \qquad (32)$$

in which the constant $q_{i,j}$ is positive.

The following $\alpha_{i,j}$ and $\dot{\Theta}_{i,j}$ are established as

$$\alpha_{i,j} = -\left(c_{i,j} + \frac{5}{2}\right)e_{i,j} - \Theta_{i,j}^T\varphi_{i,n_i}(\hat{\bar{x}}_{i,j})$$
$$+ \dot{\alpha}_{i,j} - k_{i,j}e_{i,1} \qquad (33)$$

$$\dot{\Theta}_{i,j} = \Upsilon_{i,j}(e_{i,j}\varphi_{i,j}(\hat{\bar{x}}_{i,j}) - \gamma_{i,j}\Theta_{i,j}) \qquad (34)$$

where the designed parameter $\gamma_{i,j}$ is positive.

From (30)-(34), (29) is changed as

$$\dot{V}_{i,j} \leq -\mu_i\|\varepsilon_i\|^2 - \sum_{j=1}^{n_i-1}c_{i,j}e_{i,j}^2 + \sum_{j=2}^{n_i-1}\tilde{\Theta}_{i,j}^T\gamma_{i,j}\Theta_{i,j}^T + \|\omega_i^*\|^2$$
$$- \sum_{j=1}^{n_i-1}\left(\frac{1}{\sigma_{i,j}} - \frac{1}{2q_{i,j}^2}O_{i,j}^2 - \frac{1}{2}\right)e_{\alpha_{i,h}}^2 + y_{i,d}^2 + \xi_i^{*2}$$
$$+ \frac{1}{2}\sum_{j=2}^{n_i-1}(q_{i,j}^2 + \delta_{i,j}^{*2} + \omega_{i,j}^{*2}) + \tilde{W}_{i,1}^T l_i W_{i,1}^T + \frac{1}{2}W_i^{*2}$$
$$+ \frac{1}{2}(q_{i,1}^2 + \delta_{i,1}^{*2}) + \frac{1}{2}e_{i,j+1}^2 \qquad (35)$$

**Step $i, n_i$:** Similar to [6], the following compensation signal is added:

$$\dot{\eta}_i = -\lambda_i\eta_i + u_i(t) - u_i(t - \bar{\tau}_i) \qquad (36)$$

where $\lambda_i$ is design parameter.

From (12), the error dynamic yields

$$\dot{e}_{i,n_i}(t) = u_i(t - \tau_i(t)) + \tilde{\Theta}_{i,n_i}^T\varphi_{i,n_i}(\hat{\bar{x}}_{i,n_i}) - \dot{\alpha}_{i,n_i-1}$$
$$- \lambda_i\eta_i + u_i(t) - u_i(t - \bar{\tau}_i) + k_{i,n_i}e_{i,1}$$
$$+ \Theta_{i,n_i}^T\varphi_{i,n_i}(\hat{\bar{x}}_{i,n_i}) + \delta_{i,n_i} - \omega_{i,n_i} \qquad (37)$$

The following Lyapunov function is selected:

$$V_{i,n_i} = V_{i,n_i-1} + \frac{1}{2}e_{i,n_i}^2 + \frac{1}{2\Upsilon_{i,n_i}}\tilde{\Theta}_{i,n_i}^2 + M_i$$
$$+ Q_i + N_i + G_i + \frac{1}{2}e_{\alpha_{i,n_i}}^2 \qquad (38)$$

in which $M_i = \frac{1}{2(1-\hbar_i)}\int_{t-\tau_i(t)}^t|u_i(p)|^2 dp$, $Q_i = \frac{1}{1-\hbar_i}\int_{t-\tau_i(t)}^t(\int_\vartheta^t|u_i(p)|^2 dp)d\vartheta$, $N_i = \frac{1}{2}\int_{t-\bar{\tau}_i}^t|u_i(p)|^2 dp$ and $G_i = \int_{t-\bar{\tau}_i}^t(\int_\vartheta^t|u_i(p)|^2 dp)d\vartheta$, respectively. $\Upsilon_{i,n_i} > 0$ is a constant.

Based on the Young's inequality, one obtains

$$e_{i,n_i}u_i(t-\tau_i(t)) \leq \frac{1}{2}e_{i,n_i}^2 + \frac{1}{2}|u_i(t-\tau_i(t))|^2 \qquad (39)$$

$$e_{i,n_i}u_i(t-\bar{\tau}_i) \leq \frac{1}{2}e_{i,n_i}^2 + \frac{1}{2}|u_i(t-\bar{\tau}_i)|^2 \qquad (40)$$

$$e_{i,n_i}(\delta_{i,n_i} + \omega_{i,n_i}) \leq e_{i,n_i}^2 + \frac{1}{2}\delta_{i,n_i}^{*2} + \frac{1}{2}\omega_{i,n_i}^{*2} \qquad (41)$$

$$e_{\alpha_{i,n_i}}O_{i,n_i} \leq \frac{1}{2q_{i,n_i}^2}e_{\alpha_{i,n_i}}^2 O_{i,n_i}^2 + \frac{1}{2}q_{i,n_i}^2 \qquad (42)$$

The following controller $u_i(t)$ and adaptive law $\dot{\Theta}_{i,n_i}$ are established:

$$u_i(t) = -c_{i,n_i}e_{i,n_i} - \Theta_{i,n_i}^T\varphi_{i,n_i}(\hat{\bar{x}}_{i,n_i}) - k_{i,n_i}e_{i,1}$$
$$+ \dot{\alpha}_{i,n_i} - \frac{5}{2}e_{i,n_i} + \lambda_i\eta_i \qquad (43)$$

$$\dot{\Theta}_{i,n_i} = \Upsilon_{i,n_i}(e_{i,n_i}\varphi_{i,n_i}(\hat{\bar{x}}_{i,n_i}) - \gamma_{i,n_i}\Theta_{i,n_i}) \qquad (44)$$

in which $\gamma_{i,n_i} > 0$ stands for the designed parameter.

According to (39)-(44) and the definitions of $\tilde{\Theta}_{i,j}$ and $\tilde{W}_i$, one has

$$\dot{V}_{i,n_i} \leq -\mu_i\|\varepsilon_i\|^2 - \sum_{j=1}^{n_i}c_{i,j}e_{i,j}^2 - \frac{l_i}{2}\tilde{W}_i^2 + \xi_i^{*2} - \sum_{j=1}^{n_i}\frac{\gamma_{i,j}}{2}\tilde{\Theta}_{i,j}^2$$
$$+ \left(\frac{2-\hbar_i}{2(1-\hbar_i)} + \frac{2-\hbar_i}{1-\hbar_i}\bar{\tau}_i\right)|u_i(t)|^2 + \sum_{j=1}^{n_i}\frac{\gamma_{i,j}}{2}\Theta_{i,j}^{*2}$$
$$+ \frac{1}{2}\sum_{j=2}^{n_i}(q_{i,j}^2 + \delta_{i,j}^{*2} + \omega_{i,j}^{*2}) + \left(\frac{1}{2} + \frac{l}{2}\right)W_i^{*2} + y_{i,d}^2$$
$$- \sum_{j=1}^{n_i}\left(\frac{1}{\sigma_{i,j}} - \frac{1}{2q_{i,j}^2}O_{i,j}^2 - \frac{1}{2}\right)e_{\alpha_{i,j}}^2 + \frac{1}{2}(q_{i,1}^2 + \delta_{i,1}^{*2})$$
$$- \int_{t-\bar{\tau}_i}^t|u_i(p)|^2 dp - \int_{t-\tau_i(t)}^t|u_i(p)|^2 dp + \|\omega_i^*\|^2 \qquad (45)$$

*C. Stability Analysis*

Through the definition of $Q_i$ and $G_i$, we can easily obtain

$$Q_i \leq \frac{\bar{\tau}_i}{1-\hbar_i}\int_{t-\tau_i(t)}^t|u_i(p)|^2 dp \qquad (46)$$

$$G_i \leq \bar{\tau}_i\int_{t-\bar{\tau}_i}^t|u_i(p)|^2 dp \qquad (47)$$

From (43), we can see that the design controller $u_i$ is composed of bounded signals. Hence, it fulfills $|u_i(t)| \leq \bar{u}_i$, which $\bar{u}_i > 0$ indicates a constant.

Combining (46) and (47), the inequality (45) is rewritten as

$$\dot{V}_{i,n_i} \leq - \mu_i\|\varepsilon_i\|^2 - \sum_{j=1}^{n_i} c_{i,j} e_{i,j}^2 + \xi_i^{*2} + \sum_{j=1}^{n_i} \frac{\gamma_{i,j}}{2}\Theta_{i,j}^{*2} + \|\omega_i^*\|^2$$

$$- \sum_{j=1}^{n_i}\left(\frac{1}{\sigma_{i,j}} - \frac{1}{2} - \frac{1}{2q_{i,j}^2}O_{i,j}^2\right)e_{\alpha_{i,j}}^2 + \frac{1}{2}\sum_{j=2}^{n_i}(q_{i,j}^2 + \delta_{i,j}^{*2}$$

$$+ \omega_{i,j}^{*2}) - \sum_{j=1}^{n_i}\frac{\gamma_{i,j}}{2}\tilde{\Theta}_{i,j}^2 + (\frac{l_i}{2} + \frac{1}{2})W_i^{*2} - (1 - \hbar_i)M_i$$

$$- \frac{l_i}{2}\tilde{W}_i^2 - N_i + \left(\frac{2 - \hbar_i}{2(1 - \hbar_i)} + \frac{2 - \hbar_i}{1 - \hbar_i}\bar{\tau}_i\right)\bar{u}_i^2$$

$$- \frac{1 - \hbar_i}{2\bar{\tau}_i}Q_i - \frac{1}{2\bar{\tau}_i}G_i + y_{i,d}^2 + \frac{1}{2}(q_{i,1}^2 + \delta_{i,1}^{*2}) \quad (48)$$

For the compensation signal $\eta_i$, we chose $V_{\eta_i} = \frac{1}{2}\eta_i^2$, and its derivative is

$$\dot{V}_{\eta_i} \leq -(\lambda_i - 1)\eta_i^2 + \frac{|u_i(t)|^2}{2} + \frac{|u_i(t - \tau_i)|^2}{2}$$

$$\leq -(\lambda_i - 1)\eta_i^2 + \bar{u}_i^2 \quad (49)$$

Based on the above-mentioned analysis and the decentralized controller design, there exist the following results.

*Theorem 1:* All signals can be kept to be bounded for large-scale nonlinear systems (1) with time-varying input delay, and the tracking error of each subsystem can be made arbitrarily small by utilizing the designed observer (2), adaptive laws (25)-(26), (34) and (44), virtual controllers (24) and (33) and controller (43).

*Proof:* For the whole closed-loop system, the Lyapunov function is described as

$$V = \sum_{i=1}^{N}(V_{i,n_i} + V_{\eta_i}) \quad (50)$$

The differential of the above equation is

$$\dot{V} \leq \sum_{i=1}^{N}\left\{- \mu_i\|\varepsilon_i\|^2 + \xi_i^{*2} + \frac{1}{2}(q_{i,1}^2 + \delta_{i,1}^{*2}) - \sum_{j=1}^{n_i} c_{i,j}e_{i,j}^2\right.$$

$$- \sum_{j=1}^{n_i}\left(\frac{1}{\sigma_{i,j}} - \frac{1}{2q_{i,j}^2}O_{i,j}^2 - \frac{1}{2}\right)e_{\alpha_{i,j}}^2 + \frac{1}{2}\sum_{j=2}^{n_i}(q_{i,j}^2 + \delta_{i,j}^{*2}$$

$$+ \omega_{i,j}^{*2}) + \sum_{j=1}^{n_i}\frac{\gamma_{i,j}}{2}\Theta_{i,j}^{*2} - \sum_{j=1}^{n_i}\frac{\gamma_{i,j}}{2}\tilde{\Theta}_{i,j}^2 + \frac{l_i}{2}W_i^{*2} - \frac{l_i}{2}\tilde{W}_i^2$$

$$+ \left(\frac{4 - 3\hbar_i}{2(1 - \hbar_i)} + \frac{2 - \hbar_i}{1 - \hbar_i}\bar{\tau}_i\right)\bar{u}_i^2 + \frac{1}{2}W_i^{*2} + \|\omega_i^*\|^2 + y_{i,d}^2$$

$$\left. - (1 - \hbar_i)M_i - \frac{1 - \hbar_i}{2\bar{\tau}_i}Q_i - N_i - \frac{1}{2\bar{\tau}_i}G_i - (\lambda_i - 1)\eta_i^2\right\}$$

$$\leq - CV + \aleph \quad (51)$$

where $C = \min\{C_1, C_2, \ldots, C_N\}$ and $\aleph = \sum_{i=1}^{N}\{\frac{1}{2}(q_{i,1}^2 + \delta_{i,1}^{*2}) + \frac{1}{2}\sum_{j=2}^{n_i}(q_{i,j}^2 + \delta_{i,j}^{*2} + \omega_{i,j}^{*2}) + \sum_{j=1}^{n_i}\frac{\gamma_{i,j}}{2}\Theta_{i,j}^{*2} + y_{i,d}^2 + (\frac{1}{2} + \frac{l_i}{2})W_i^{*2} + \|\omega_i^*\|^2 + \left(\frac{4-3\hbar_i}{2(1-\hbar_i)} + \frac{2-\hbar_i}{1-\hbar_i}\bar{\tau}_i\right)\bar{u}_i^2 + \xi_i^{*2} + \frac{1}{2}(q_{i,1}^2 + \delta_{i,1}^{*2})\}$. One defines $C_i = \min\{\mu_i/\lambda_{\min}(P_i), 2c_{i,j}, 2(\frac{1}{\sigma_{i,j}} - \frac{1}{2q_{i,j}^2}O_{i,j}^2 - \frac{1}{2}), \gamma_{i,j}\Upsilon_{i,j}, l_i\zeta_{i,1}, (1 - \hbar_i), \frac{1-\hbar_i}{2\bar{\tau}_i}, 1, -\frac{1}{2\bar{\tau}_i}, 2(\lambda_i - 1)\}$. ∎

## IV. SIMULATION RESULTS

The presented control strategy is applied in the two inverted pendulum systems which are connected through a spring with $\theta_1 = x_{1,1}$, $\theta_2 = x_{2,1}$, $\dot{\theta}_1 = x_{1,2}$ and $\dot{\theta}_2 = x_{2,2}$, and the systems are modeled by the following form [11]:

$$\begin{cases} \dot{x}_{1,1} = x_{1,2} \\ \dot{x}_{1,2} = \frac{u_1(t - \tau_1(t))}{J_1} + \frac{kr^2}{4J_1}\sin(x_{2,1}) + \frac{kr}{2J_1}(l - b) \\ \quad + \left(\frac{m_1 gr}{J_1} - \frac{kr^2}{4J_1}\right)\sin(x_{1,1}) \\ y_1 = x_{1,1} \end{cases} \quad (52)$$

$$\begin{cases} \dot{x}_{2,1} = x_{2,2} \\ \dot{x}_{2,2} = \frac{u_2(t - \tau_2(t))}{J_2} + \frac{kr^2}{4J_2}\sin(x_{1,1}) + \frac{kr}{2J_2}(l - b) \\ \quad + \left(\frac{m_2 gr}{J_2} - \frac{kr^2}{4J_2}\right)\sin(x_{2,1}) \\ y_2 = x_{2,1} \end{cases} \quad (53)$$

where the pendulum end masses are defined as $m_1$ and $m_2$. $k$ and $l$ denote spring constant and the natural length of the spring, respectively. $J_1$ and $J_2$ represent the moments of inertia. The distance between the pendulum hinges, gravitational acceleration and pendulum height are indicated as $b$, $g$ and $r$, respectively. The reference signal is given as $y_{i,d} = \sin(t)$. The time-varying input delay is expressed as $\tau_i(t) = 0.005\sin(t) + 0.01$. The initial values and parameters of the systems are described in Table I.

TABLE I
THE MODEL PARAMETERS

| | | | |
|---|---|---|---|
| $x_{1,1}(0) = -0.04$ | $x_{1,2}(0) = 0.02$ | $x_{2,1}(0) = -0.04$ | $x_{2,2}(0) = 0.02$ |
| $\hat{x}_{1,1}(0) = -0.04$ | $\hat{x}_{1,2}(0) = 0.02$ | $\hat{x}_{2,1}(0) = -0.04$ | $\hat{x}_{2,2}(0) = 0.02$ |
| $\eta_1(0) = 10$ | $\eta_2(0) = 10$ | $\hat{\Theta}_{1,2}(0) = 35$ | $\hat{\Theta}_{2,2}(0) = 49.9$ |
| $c_{1,1} = 500$ | $c_{1,2} = 0.1$ | $c_{2,1} = 500$ | $c_{2,2} = 0.1$ |
| $m_1 = 2$ kg | $m_2 = 2.5$ kg | $J_1 = 5$ kg | $J_2 = 6.25$ kg |
| $k = 100$ N/m | $r = 0.5$ m | $l = 0.5$ m | $b = 0.5$ m |
| $g = 9.81$ m/s$^2$ | $\sigma_1 = 10$ | $\sigma_2 = 10$ | $\lambda_1 = 70$ |
| $\lambda_2 = 70$ | $k_{1,1} = 1$ | $k_{1,2} = 20$ | $k_{2,1} = 1$ |
| $k_{2,2} = 5$ | $\Upsilon_{1,2} = 0.5$ | $\Upsilon_{2,2} = 0.01$ | $\gamma_{1,2} = 2$ |
| $\gamma_{2,2} = 1$ | | | |

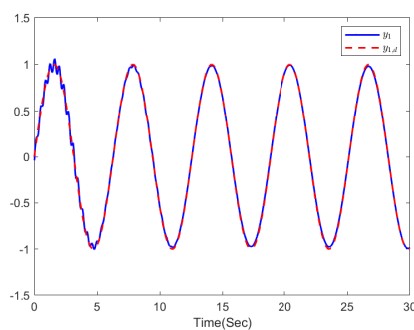

Fig. 1.   Curves of $y_1$ and $y_{1,d}$.

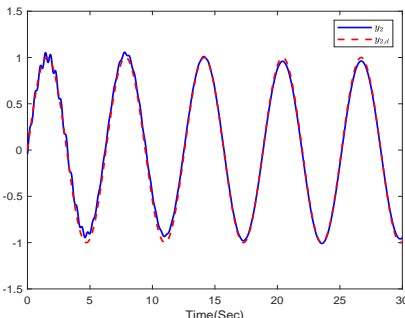

Fig. 2.   Curves of $y_2$ and $y_{2,d}$.

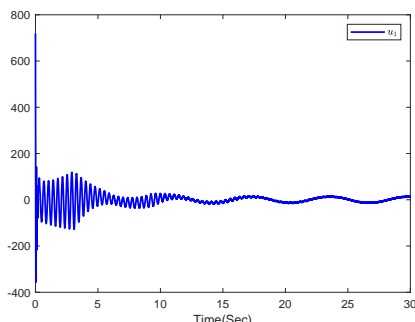

Fig. 3.   Curve of $u_1$ with input-delay $\tau_1(t) = 0.005\sin(t) + 0.01$.

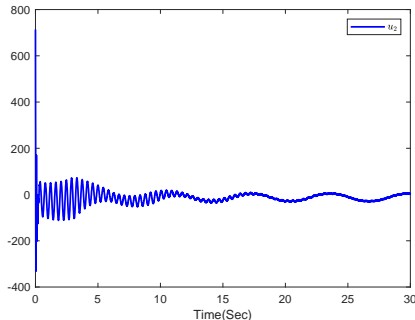

Fig. 4.   Curve of $u_2$ with input-delay $\tau_2(t) = 0.005\sin(t) + 0.01$.

Through the designed adaptive laws (25), (26) and (44), real controller (43) and virtual controller (24) to control two inverted pendulum systems, Figs. 1-4 display the results of the simulation. Fig. 1 and Fig. 2 depict the good tracking performance of the control outputs of each subsystem on the ideal reference trajectory. Fig. 3 and Fig. 4 plot the trajectory of the control input $u_i$ with $\tau_i(t) = 0.005\sin(t) + 0.01$. In summary, the simulation results verify the availability of the proposed strategy.

## V. CONCLUSION

Observer-based fuzzy adaptive decentralized control strategy has been studied for nonlinear interconnected systems against input delay. By introducing the tracking error into the designed of the state observer, the unmeasurable states have been estimated. Moreover, the compensation signal has been prompted to compensate for the impact of input delay. The "explosion of complexity" problem have been overcome by combining DSC and first-order filters. It has displayed that the presented control strategy guarantees that all variables of whole system are semiglobally uniformly ultimately bounded. Finally, the inverted pendulum systems has been applied to clarify that the proposed method is valid. In the future work, we will strive to apply the presented method in this paper to other systems [12].

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
