# OpenReview forum: "Observer-Based Adaptive Decentralized Control for Interconnected Nonlinear Systems with Input Delay"
_IEEE.org/ICIST/2024/Conference — IEEE ICIST 2024 Conference Submission_

### Official Review · Reviewer_knrv · 2024-08-22
**Accept**

**Rating:** 7
**Confidence:** 3

**Review:**

In this paper, " Observer-Based Adaptive Decentralized Control for Interconnected Nonlinear Systems with Input Delay", an observer-based fuzzy adaptive decentralized control strategy is proposed. Firstly, a tracking error-based state observer is proposed to estimate the unknown state of interconnected systems. Then, a compensating signal is added to handle the time-varying input delay. The article has clear logic and organization, but there are still some problems. My specific feedback is as follows :1) In the introduction section, the interconnection system and input delay are not introduced enough. 2) What are the advantages of tracking error-based observers compared with traditional state observers?

---

### Official Review · Reviewer_WMBp · 2024-08-22
**This article is quite fascinating and of high quality.**

**Rating:** 7
**Confidence:** 3

**Review:**

The paper titled "Observer-Based Adaptive Decentralized Control for Interconnected Nonlinear Systems with Input Delay" design the observer-based fuzzy adaptive decentralized control strategy. Firstly, a novel state observer based on tracking error is proposed to estimate unknown states for the interconnected systems. A compensation signal is added to handle the time-varying input delay. Then, the dynamic surface control techniques are utilized in the backstepping approach to overcome the “explosion of complex” problem. Meanwhile, through designed schemes, all variables in the closed-loop systems are promised to be bounded. The errors converge into bounded compact set around the origin. Finally, the effectiveness of the designed scheme is validated via a simulation example. My specific feedback is as follows: 1) The advantages of the author's method compared to other methods are not stated in the contribution. 2) Some formatting issues need to be addressed.

---

### Official Review · Reviewer_JN6Q · 2024-08-22
**This article is very interesting and a good one**

**Rating:** 7
**Confidence:** 3

**Review:**

In this paper, an observer-based fuzzy adaptive decentralized control strategy was proposed for nonlinear interconnected systems against input delay. The obtained result is valuable and can be accepted if the following problems can be clarified.
(1) In the introduction, the shortages of those relevant studies are suggested to be further summarized.
(2) The definition of Lemma 1 should be provided in detail, including each variable
 (3) In the end of Section 1, the organization of this study is suggested to be summarized.
(4) There exist several spelling and grammar errors. Please check carefully and further polish
 (5) In the simulation section, more analysis can be added to better explain the main results of this paper, that's not enough.

---

### Comment · Reviewer_JN6Q · 2024-08-21
**This article is very interesting and a good one**

In this paper, an observer-based fuzzy adaptive decentralized control strategy was proposed for nonlinear interconnected systems against input delay. The obtained result is valuable and can be accepted if the following problems can be clarified.
(1)	In the introduction, the shortages of those relevant studies are suggested to be further summarized.
(2)	The definition of Lemma 1 should be provided in detail, including each variable
(3)	In the end of Section 1, the organization of this study is suggested to be summarized.
(4)	There exist several spelling and grammar errors. Please check carefully and further polish
(5)	In the simulation section, more analysis can be added to better explain the main results of this paper, that's not enough.

---

### Decision · Program_Chairs · 2024-09-06

Accept (Oral)